# Complexity of active medicinal ingredients in radix scutellariae with sodium hydrosulfite exposure

Ying Shen[1], Wei Cong[1,2], Ai-hua Zhang[1], Xiangcai Meng[1]*

**1** Department of Pharmacognosy, Heilongjiang University of Chinese Medicine, Harbin, China, **2** GAP Research Center, Heilongjiang University of Chinese Medicine, Harbin, China

* 13845044491@163.com

**Data Availability Statement:** All relevant data are within the paper and its Supporting Information files.

## Abstract

Both plants and animals are living things made up of similar cells as well as organelles, and their essence of life is the same. However, plants face more environmental stress than animals and generate excessive reactive oxygen species (ROS), a group of small molecules that can harm proteins, necessitating distinctive metabolic processes. Secondary metabolites in plants are a group of chemical components that can eliminate ROS and can also exhibit medicinal properties; therefore, herbal medicines are often closely linked to the ecological significance of secondary metabolites. Why plants contain so many, not few, active medicinal ingredients is unknown. The root of *Scutellaria baicalensis*, a popular herbal medicine, is rich in various flavonoids with diverse structural features. Sodium hydrosulfite ($Na_2S_2O_4$) can produce $O^-_2$ radicals and induce physical conditions under environmental stress. Using UHPLC-ESI-Q-TOF-MS/MS analysis, a total of 25 different compounds were identified in the roots of *S. baicalensis* between the $Na_2S_2O_4$ groups and suitable conditions. Based on the results of the t-test (P<0.05) performed for the groups and ions with values of VIP $\geq$ 2, the most significantly different chemical markers with $Na_2S_2O_4$ treatment were shikimic acid, citric acid, baicalin, wogonoside, baicalein, wogonin, 3,5,7,2',6'-pentahydroxyflavanone, 5,2',6'-trihydroxy-7,8-dimethoxy flavone, chrysin, eriodictyol, 5,8-dihydroxy-6,7-dimethoxy flavone, skullcapflavone II, and 5,7-dihydroxy-6,8,2',3'-tetrame thoxyflavone, and most of them were free flavonoids with many phenolic hydroxyl or methoxyl groups and characteristically high antioxidant activities. *S. baicalensis* roots modified their ability to eliminate ROS and maintained the equilibrium of ROS through the multitudinous biosynthesis and conversion of flavonoids, which is similar to the equilibrium established by an intricate buffer solution and perfectly explains the diversity and complexity of medicinal plant ingredients.

## Introduction

Animals can avoid unfavourable circumstances, and the survival of native species is mainly dependent on intraspecific or interspecific competition for foods, not the ecological

**Funding:** The authors wish to thank for providing financial supports from the National Science Foundation (81573523).

**Competing interests:** The authors have declared that no competing interests exist.

environment; however, immobile plants can get enough light from the sun, but must face high temperatures, drought, and low soil fertility stress factors, so various adverse effects are almost uninterrupted. Under stress, the light energy absorbed by chloroplasts far outweighs the need to capture $CO_2$, leading to a surfeit of light energy. In addition, closed stomata under stress block $O_2$ emissions outward, and the reduction of $O_2$ to $O^{\bullet-}_2$ (Mehler reaction) is exacerbated [1,2]. $O^{\bullet-}_2$ radicals can be converted further into $\cdot OH$ and $H_2O_2$. These molecules have a strong oxidation power and are described as reactive oxygen species (ROS) that can modify the structures of proteins, including enzymes, by affecting disulfide bridges and regulating various metabolic processes. Therefore, a suitable level of ROS also acts as an indispensable messenger to regulate various physiological actions in plants [3,4], but once ROS are overproduced, a range of destructive forces follow, such as altered adjacent molecular configurations, a reduced cell-membrane stability, DNA strand destruction, protein crosslinking, and peptide chain breakage, resulting in metabolic disorders and even cell death [5]. It has been confirmed that increased ROS levels are a result of stress, resulting in a 3-fold increase in $O^-_2$ and a 10-fold increase in $H_2O_2$ under certain conditions [6]. Under high levels of $O^-_2$, metabolic alterations under stress can be reproduced; the activities of superoxide dismutase (SOD), catalase (CAT), and peroxidase (POD) are decreased, and the content of baicalein, an antioxidant compound, was shown to increase from 0.28% to 1.96% [7].

ROS are eliminated mainly by antioxidant enzymes, including SOD, CAT, and POD, and secondary metabolites, including phenolic compounds, carotenoids, and tocopherols. $O^-_2$ is converted to $H_2O_2$ either spontaneously or by SOD and then converted into $H_2O$ and $O_2$ by CAT or POD. A suitable level of ROS is an indispensable part of metabolism; however, antioxidant enzymes are also proteins. In particular, some -SH groups that maintain the secondary structure and tertiary structure of the enzymes are liable to be injured if an undue amount of ROS in plants is generated under severe abiotic stresses [5]. Therefore, this might be the reason why plants possess the ability to produce secondary metabolites in addition to antioxidant enzymes.

The secondary metabolites in plants are numerous; by ultra-high-performance liquid chromatography, a total of 132 metabolites in *Scutellaria baicalensis*, 447 metabolites in *Isatis indigotica* Fortune, 122 in *Moringa oleifera* leaves, and 128 in American ginseng roots were identified [8–11], most of which were secondary metabolites. It has been proven that the proportion of secondary metabolites varies according to ever-changing environmental conditions [12]. It is known that the ingredients in herbal medicines are complex, so it is impossible to assess their value by only one or several secondary metabolites. These compounds are usually put into classes with similar structures, whose changes result in variations in their activities [13]. Secondary metabolites are important for plants to adapt to adversity; the reason why plants contain so many, not few, secondary metabolites is unknown. Additionally, the connection between the secondary metabolites is also unknown. *S. baicalensis* Georgi is distributed throughout semi-arid (steppe) climates and often undergoes severe drought stresses, which are a main factor affecting the biosynthesis of flavonoids [14,15]. To protect themselves under adverse environmental conditions, plants produce various kinds of secondary metabolites, which are also active ingredients exhibiting anti-inflammatory, antitumour, and anti-HIV activities [16].

Metabolomics can acquire comprehensive information about various metabolites by using untargeted biochemical approaches to monitor metabolites and is a new technique used in the field of plant research. Quantitative plant metabolomics, with the ability to improve the comprehensive understanding of plant metabolism under different conditions [17], has been considered the most promising approach for the detection of primary and secondary stress-response metabolites [18]. Since ROS have a dual effect, too much or too little is harmful, and

plants require that the equilibrium of ROS is exactly maintained. The large quantities and constantly changing activities of secondary metabolites are perhaps an important tactic for plants to respond to changing environments as quickly and as delicately as possible, which is probably performed through the biosynthesis and conversion of numerous flavonoids for *S. baicalensis*. $O^-_2$ is an original ROS that is not present long term in nature due to its poor stability, with a half-life of only approx. 1 μs. Sodium hydrosulfite ($Na_2S_2O_4$) can produce $O^-_2$ spontaneously under alkaline conditions [19] and is regarded as an important carrier [20]; $Na_2S_2O_4$ can easily control $O^-_2$ production, creating stress physiology. Here, $Na_2S_2O_4$ was employed to induce stress, and we investigated the different secondary metabolites, the biological significance of flavonoids in *S. baicalensis* under stress and the diversity and complexity of medicinal ingredients.

## Materials and methods

### Plant materials and reagents

*S. baicalensis* Georgi samples were collected from the medical plant garden at Heilongjiang University of Chinese Medicine. In October 2018, 2-year-old plants of *S. baicalensis* were sprayed with aqueous $Na_2S_2O_4$ aqueous solutions with concentrations of 40 μmol/L, 0.4 μmol/L, and 0.004 μmol/L, and the control group was kept moist with just water. The samples were collected 0, 1, 2, and 3 days after spraying. A total of 5 plants were selected for each sample, the dirt was washed off, the xylem was removed, and the samples were freeze-dried and then ground into a fine powder.

Methanol (HPLC-grade) was purchased from Fisher Scientific Corporation (Loughborough, UK); HPLC grade acetonitrile was obtained from Merck (Darmstadt, Germany); leucine-enkephalin was purchased from SIGMA Corporation (USA); ultrapure water was produced by a Milli-Q Ultra-pure water purification system (Millipore Corporation, MA, USA). All other reagents were of analytical grade.

### Preparation of extracts for UHPLC-ESI-Q-TOF-MS/MS analysis

Collection and preparation of plant samples: Fine powder (150 mg) and 50 mL 70% methanol were placed into conical flasks and ultrasonically extracted for 1 h, and the volume lost was replaced with fresh 70% methanol. Finally, the supernatant was filtered with a 0.22 μm microporous filter for UPLC analysis.

### Analytical conditions

Ultra-performance liquid chromatography: Chromatographic separation was performed on an ACQUITY UPLC system (Waters Corporation, Milford, MA) consisting of a binary solvent system, a sample manager and a column compartment. The column used was a UPLC$^{TM}$ BEH $C_{25}$ column (100 mm× 2.1 mm 1.8 μm, Waters Corporation, Milford, USA). The column temperature was maintained at 40°C for all analyses, and the autosampler temperature was maintained at 10°C. The optimal mobile phase consisted of a linear gradient system of (A) 0.1% formic acid in acetonitrile and (B) 0.1% formic acid in water: 0 to 1.5 min, 16 to 22% A; 1.5 to 5 min, 22 to 30% A; 5 to 9 min, 30 to 40% A; 9 to 12 min, 40 to 70% A; and 12 to 15 min, 70 to 100% A. The detection wavelengths were those used in a full ultraviolet wavelength scan from 190 to 400 nm. The flow rate was set to 0.4 mL/min. The injection volume was 3 μL. The detection of positive and negative ions was performed by the continuous flow of the analytes to the mass spectrometer. All the samples were kept at 4°C during the analysis.

## Mass spectrometry

Positive ionization mode: The capillary voltage was 3.0 kV, the sampling cone voltage was 25 V, the extraction voltage was 4.0 V, the desolvation gas temperature was 350˚C, the desolvation gas flow was 600 L/h, the source temperature was 110˚C, and leucine enkephalin at a concentration of 0.2 ng/mL was used via a lock spray interface and introduced with a flow rate of 100 μL/min for monitoring in positive ionization mode ([M+H]+ = 556.2771) to ensure accuracy during the MS analysis. The lock spray frequency was set to 5 s, and scan averaging for the correction was performed every 0.02 s and 0.4 s. The scanning range was $m/z$ 100~1500.

Negative ionization mode: the capillary voltage was 2.2 kV, the sampling cone voltage was 25 V, the extraction voltage was 3.5 V, the desolvation gas temperature was 350˚C, the desolvation gas flow was 600 L/h, the source temperature was 110˚C, and leucine enkephalin at a concentration of 0.2 ng/mL was used via a lock spray interface at a flowrate of 100 μL/min for monitoring in negative ionization mode ([M+H]+ = 556.2771) to ensure accuracy during the MS analysis. The lock spray frequency was set to 5 s, and scan averaging for correction was performed every 0.02 s and 0.4 s per scan. The scanning range was $m/z$ 100~1500.

## Results

### Identification of chemical markers in radix scutellariae

MassLynx V 4.1 was employed for the analysis of the chemical constituents of Radix Scutellariae. The chemical composition was elucidated by the spectral information obtained from secondary ion mass spectrometry, which was cross-referenced with the retention time, mass-to-charge ratio, molecular weight, structural formula and elemental composition of known ingredients in Radix Scutellariae. Based on the VIP results, the different candidate ions observed in the $Na_2S_2O_4$ treatment groups and the control group were tentatively identified. When taking the identification of baicalin as an example, in positive mode, the ion (RT = 4.92 min and [M +H]$^+$ = 447.12) detected in the $Na_2S_2O_4$-treated sample was calculated to be $C_{21}H_{19}O_{11}$ based on the elemental composition, fractional isotope abundance, and Chemspider database information. The main MS/MS fragment ions of peak 9 were $m/z$ 271 and $m/z$ 253, indicating that the fragments may be $C_{15}H_{11}O_5^-$ and $C_{15}H_9O_4^-$, which could indicate the loss of glucuronic acid (176 Da) and $H_2O_2$ (25 Da), respectively. With this integrated information, the ion was finally confirmed to be that of baicalin (Fig 1). The corresponding mass spectra and related structures are shown in Fig 1. According to the above-mentioned analytical method and relevant literature [8,21], a total of 25 chemical markers that were differentially expressed between the two Radix Scutellariae treatment groups was successfully identified, including 19 candidate ions in positive ion mode and 6 candidate ions in negative mode. Using Waters Masslynx software, we finally confirmed their identities with the MS/MS data. The UPLC-HDMS

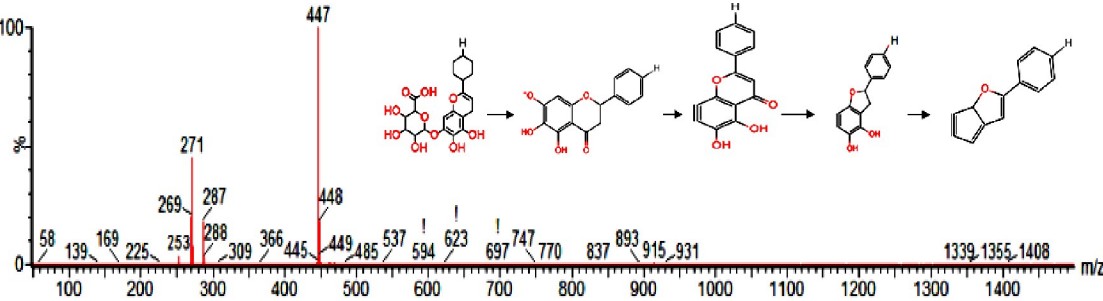

**Fig 1. MS/MS spectrum and the proposed fragmentation pathway of baicalin.**

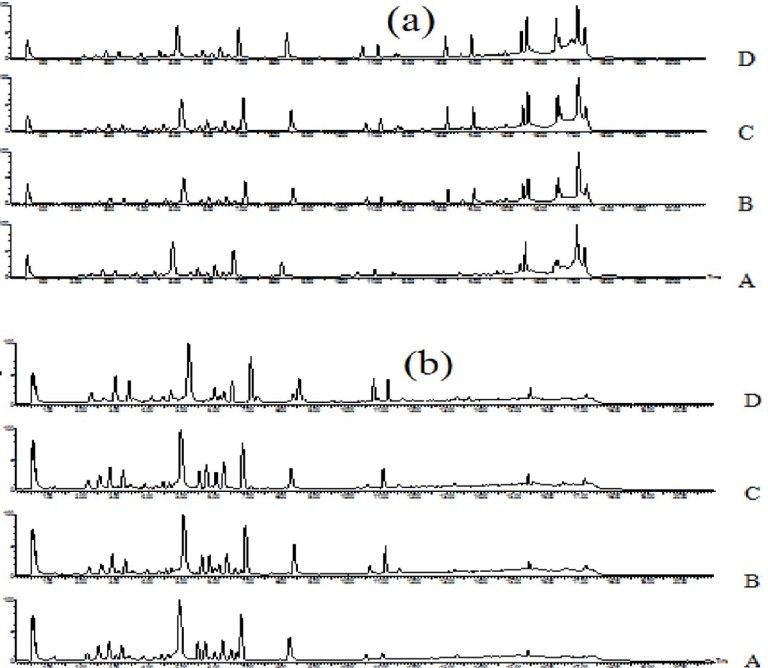

**Fig 2. UPLC chromatograms of Scutellaria root extracts analysed in positive ionization mode and negative ionization mode.** (a) Positive ion mode; (b) Negative ion mode. A: 0 days; B: 1 day; C: 2 days; D: 3 days; E: 4 days.

chromatograms of Scutellaria root (Radix Scutellariae) in positive ion mode and in negative ion mode are shown in Fig 2. Detailed information on all the identified components in different treatments are shown in Table 1.

With aqueous $Na_2S_2O_4$ at different concentrations, citric acids presented a downtrend, and the other components exhibited an uptrend, as shown in Fig 3.

## Characteristic multivariate metabolomic data analysis

A PCA model was used to identify the difference in the metabolites of the $Na_2S_2O_4$ treatment and control groups. The PCA score plots are shown in Fig 4. OPLS-DA was used to discriminate between the groups as well. As shown in Fig 5, the group differences were clearly divided into three regions, indicating that there were significant chemical differences between them, and the established metabolomics method could successfully characterize the chemical characteristics. VIP values are commonly used to evaluate the contribution of variables in OPLS-DA. Based on the results of the t-test ($P<0.05$) performed for the different groups, ions with VIP values $\geq 2$ were selected and regarded as the most significantly different chemical markers of the $Na_2S_2O_4$ treatment and control groups. Therefore, shikimic acid, citric acid, baicalin, wogonoside, baicalein, wogonin, 3,5,7,2',6'-pentahydroxyflavanone, 5,2',6'-trihydroxy-7,8-dimethoxy flavone, chrysin, eriodictyol, 5,8-dihydroxy-6,7- dimethoxyflavone, skullcapflavone II, and 5,7-dihydroxy-6,8,2',3'-tetramethoxy flavone presented significant differences.

## Discussion

### Secondary metabolism was enhanced

In this paper, only two primary metabolites were observed in the mass spectral data. Citric acid, a major substance in the tricarboxylic acid cycle, decreased slightly and continuously

**Table 1. Characterization of compounds in fresh Scutellaria root extracts by UPLC-HDMS.**

| Peak No. | Rt (min) | Selected ion | Measured mass (m/z) | Calc. mass m/z | Error (ppm) | Formula | MS/MS fragment ion (m/z) | Identification |
|---|---|---|---|---|---|---|---|---|
| 1 | 0.54 | [M-H]⁻ | 173.15 | 173.03 | 3.3 | $C_7H_{11}O_5$ | 155,137,111 | Shikimic Acid |
| 2 | 0.62 | [M-H]⁻ | 191.14 | 191.15 | 2.9 | $C_6H_7O_7$ | 173,147 | Citric acid |
| 3 | 2.17 | [M+H]⁺ | 305.26 | 305.21 | 4.0 | $C_{15}H_{13}O_7$ | 287,153 | 3,5,7,2',6'-Pentahydroxyflavanone |
| 4 | 2.8 | [M+H]⁺ | 549.15 | 549.16 | 1.6 | $C_{26}H_{29}O_{13}$ | 531,513,495411,375 | Aspenin-6-C-Arabinose-8-C-glucoside or Chrysin-6-C-glucose-8-C-Araboside |
| 5 | 3.19 | [M+H]⁺ | 549.15 | 549.16 | 0.9 | $C_{26}H_{29}O_{13}$ | 531,513,495411,375 | Aspenin-6-C-Arabinose-8-C-glucoside or Chrysin-6-C-glucose-8-C-Araboside |
| 6 | 4.16 | [M+H]⁺ | 477.10 | 477.10 | 2.1 | $C_{22}H_{21}O_{12}$ | 301,286 | 5,7,2'-trihydroxy-6-methoxyflavonoid-7-O-glucuronide |
| 7 | 4.35 | [M-H]⁻ | 287.26 | 287.19 | 2.6 | $C_{15}H_{11}O_6$ | 251,135 | Eriodictyol |
| 8 | 4.56 | [M+H]⁺ | 347.07 | 347.07 | 4 | $C_{17}H_{15}O_8$ | 332,314 | 5,7,2',5'- tetrahydroxy -8,6'- dimethyl oxyflavone |
| 9 | 4.92 | [M+H]⁺ | 447.12 | 447.09 | 2.5 | $C_{21}H_{19}O_{11}$ | 271,253 | Baicalin |
| 10 | 5.68 | [M+H]⁺ | 447.09 | 447.09 | 4 | $C_{21}H_{19}O_{11}$ | 285 | Oroxylin A-5-O glucoside |
| 11 | 5.83 | [M+H]⁺ | 447.09 | 447.09 | 4.7 | $C_{21}H_{19}O_{11}$ | 271 | Baicalin isomers |
| 12 | 5.95 | [M+H]⁺ | 477.10 | 477.10 | 4.2 | $C_{22}H_{21}O_{12}$ | 301,286 | 5,7,8-trihydroxy-6-methoxyflavone-7-O-glucuronide |
| 13 | 6.19 | [M+H]⁺ | 461.10 | 461.10 | 4.6 | $C_{22}H_{21}O_{11}$ | 285,270 | Wogonoside |
| 14 | 6.2 | [M-H]⁻ | 429.08 | 429.08 | 3.5 | $C_{21}H_{17}O_{10}$ | 253 | Chrysin -7-O-glucuronide |
| 15 | 6.44 | [M+H]⁺ | 477.10 | 477.10 | 2.9 | $C_{22}H_{21}O_{12}$ | 301 | 5,6,7-trihydroxy-8-methoxyflavone-7-O-glucuronide |
| 16 | 7.01 | [M+H]⁺ | 315.30 | 315.22 | 1.1 | $C_{17}H_{15}O_6$ | 282,285 | 5,8-dihydroxy-6,7-dimethoxyflavones |
| 17 | 7.05 | [M-H]⁻ | 489.09 | 489.10 | 0.8 | $C_{23}H_{21}O_{12}$ | 313,298 | 5,7-dihydroxy-6,8-dimethoxyflavone-7-O-glucuronide |
| 18 | 7.97 | [M-H]⁻ | 329.88 | 329.85 | 3.5 | $C_{17}H_{13}O_7$ | 299 | 5,2',6'-Trihydroxy-7,8-dimethoxyflavone |
| 19 | 8.03 | [M+H]⁺ | 301.06 | 301.07 | 3.7 | $C_{16}H_{13}O_6$ | 286 | tenaxin II |
| 20 | 8.21 | [M+H]⁺ | 271.05 | 271.06 | 3.6 | $C_{15}H_{11}O_5$ | 253,241 | Baicalein |
| 21 | 10.51 | [M+H]⁺ | 285.05 | 285.07 | 4.6 | $C_{16}H_{13}O_5$ | 270 | Wogonin |
| 22 | 10.67 | [M+H]⁺ | 255.06 | 255.06 | 4.5 | $C_{15}H_{11}O_4$ | 209 | Chrysin |
| 23 | 11.01 | [M+H]⁺ | 375.10 | 375.18 | 4.3 | $C_{19}H_{19}O_8$ | 360,345,327 | Skullcapflavone II |
| 24 | 11.05 | [M+H]⁺ | 375 | 375 | 1.1 | $C_{19}H_{19}O_8$ | 345 | 5,7-dihydroxy-6,8,2',3'-tetramethoxyflavone |
| 25 | 11.45 | [M+H]⁺ | 345.0 | 345.09 | 4.3 | $C_{25}H_{17}O_7$ | 330,315 | 5,2-dihydroxy-6,7,8-trimethoxyflavones |

with increased $Na_2S_2O_4$ exposure, indicating that its primary metabolism was weakened. Shikimic acid, from which various flavonoids originate, is a branch point of the primary and the secondary metabolic pathways, and the decreased citric acid content and increased shikimic acid content indicated enhanced secondary metabolism, resulting in increased secondary metabolite production; additionally, the content of a total of 13 secondary metabolites increased with the $Na_2S_2O_4$ treatment. Shikimic acid is located upstream of citric acid, which indicates that more shikimic acid would be converted into secondary metabolites.

## Biological significance of varied compounds

Three features of these enhanced secondary metabolites became prominent. First, the molecular structure of the secondary metabolites dictated the biological effect based on the number and sites of the phenolic hydroxyl groups in flavonoids. Flavonoids are a series of compounds containing two benzene rings connected by 3 carbon atoms, and phenolic hydroxyls are a regular moiety on them, as shown in Fig 6. It has been proven that the number of hydroxyl groups on the B ring directly impacts the activity, which is also markedly enhanced when a double

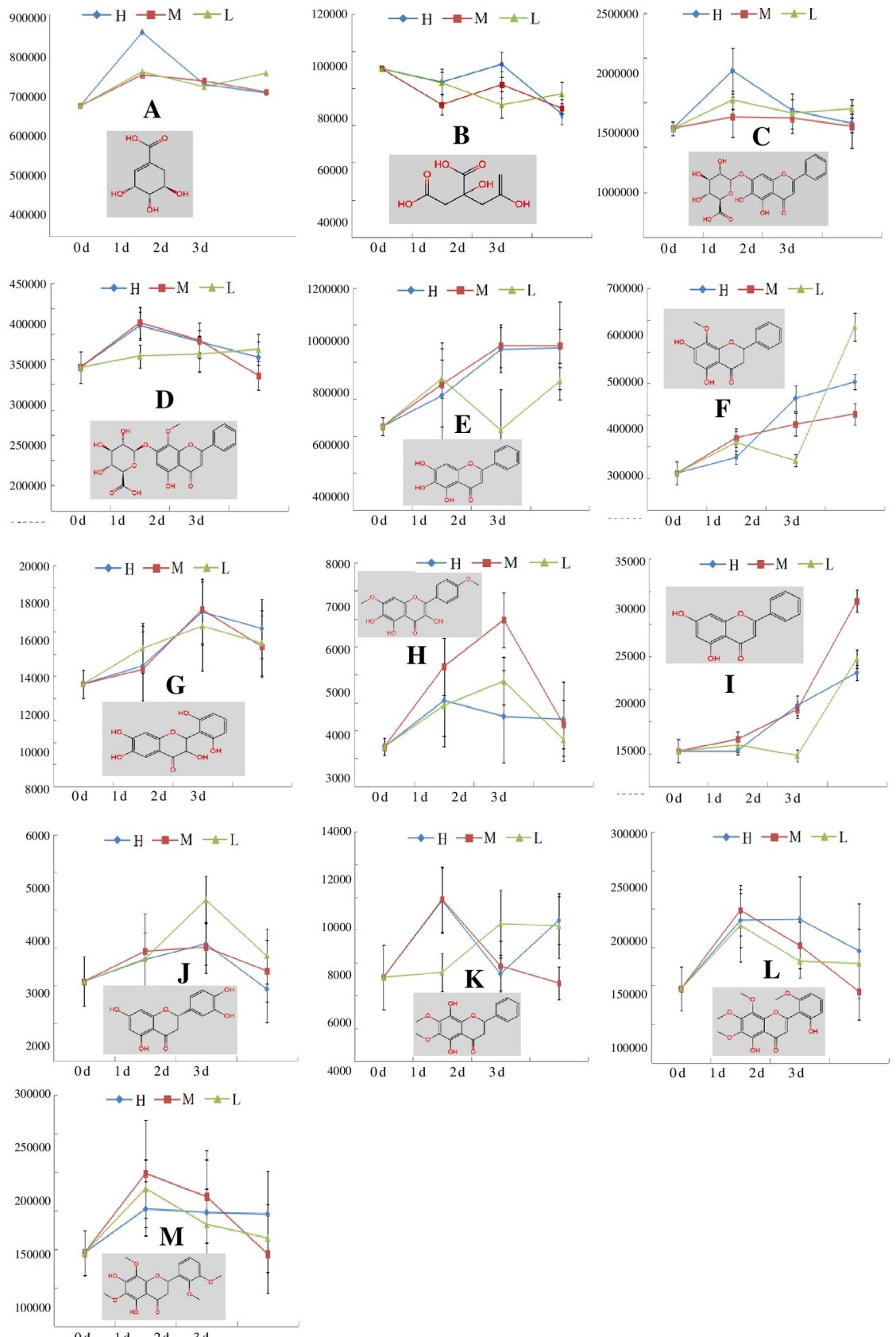

**Fig 3. Changes in the contents of 13 compounds under Na₂S₂O₄ treatment.** A: shikimic acid; B: citric acid; C: baicalin; D: wogonoside; E: baicalein; F: wogonin; G: 3,5,7,2',6'-pentahydroxyflavanone; H: 5,2',6'-trihydroxy-7,8-dimethoxyflavone; I: chrysin; J: eriodictyol; K: 5,8-dihydroxy-6,7-dimethoxyflavone; L: Skullcapflavone II; M: 5,7-dihydroxy-6,8,2',3'-tetramethoxyflavone; L: 0.004 μmol/L Na₂S₂O₄; M: 0.4 μmol/L Na₂S₂O₄; H: 40 μmol/L Na₂S₂O₄.

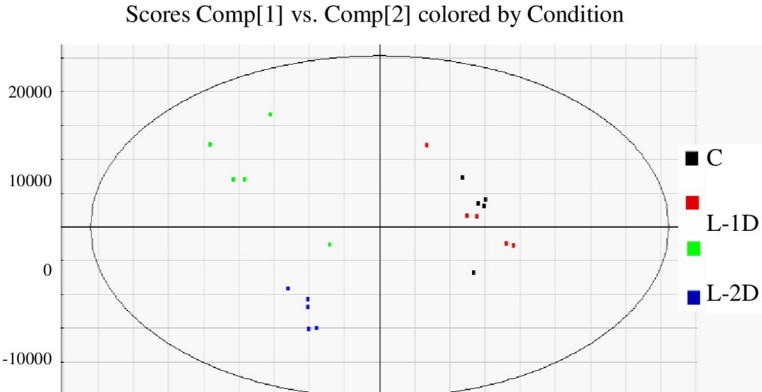

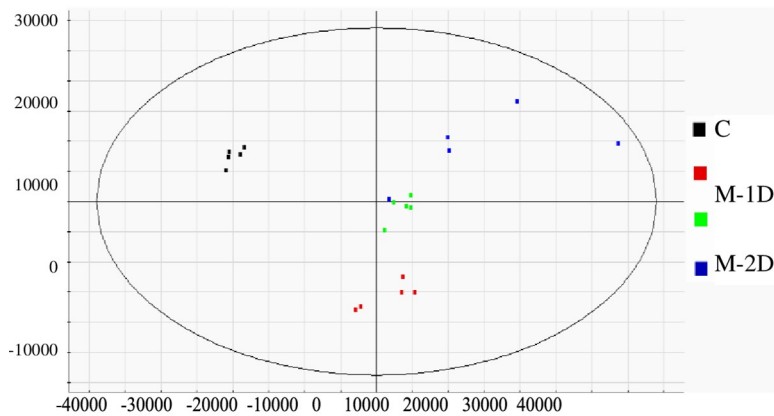

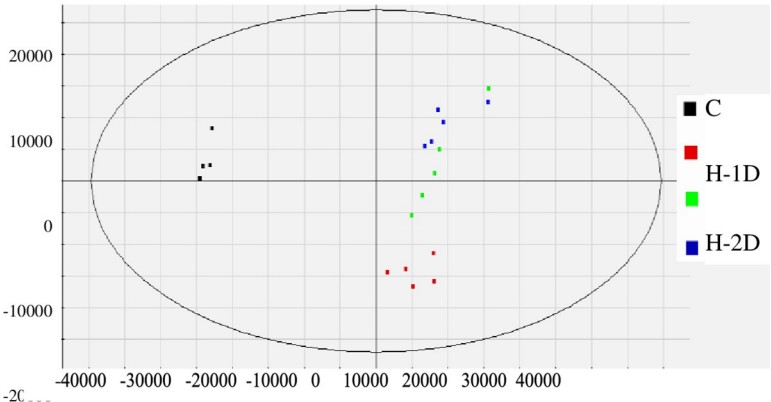

**Fig 4. Score plots of UPLC-Q/TOF-MS data generated via PCA in positive ion mode.**

Scores Comp[1] vs. Comp[2] colored by Condition

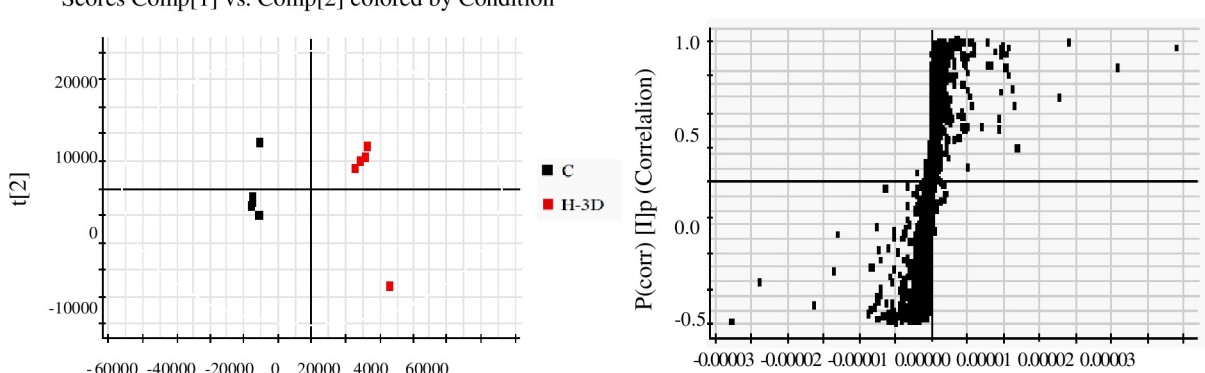

**Fig 5. Combination of S-score values of OPLS-DA detected in positive ion mode for the low-$Na_2S_2O_4$ treatment sample and the control sample on the third day.**

bond is introduced into the C ring [22]. The hydroxyl group at positions C-5 and C-7 together in the A ring, as well as the C-3', C-4' and C-5' sites on the B ring, can all increase the activities obviously [23,24]. A hydroxyl group can also be located in another position, such as C-6 [25]. It has been proven that baicalein is 2~5 times more antibiotically active than baicalin and 1~3 times better than baicalin at inhibiting IL-1β and converting enzymes [24]. Second, except for baicalin and wogonoside, the other flavonoids were free flavonoids without sugar moieties. The biosynthesis of flavonoids is performed in the endoplasmic reticulum, and it is difficult for flavonoid glycosides with hydrophilic sugar moieties to freely pass through biomembranes with a lipophilic phospholipid bilayer. The biomembranes of animals and plants are the same; therefore, the biological effect of glycosides would be limited in animals and plants unless the sugar moieties were removed. A study showed that the activities of flavanone disappear when a sugar moiety is introduced into the A ring [22], and another study showed that baicalein is more than 7 times more bioavailable than baicalin due to hydrophilic variations [26]. Flavonoid glycosides therefore are regarded as superfluous flavonoids; when required, they work mainly after their conversion into free flavonoids [27–29]. Third, the contents of some components with methoxy groups increased under high levels of ROS. The methoxy groups on flavonoids have the ability to donate electrons and exhibit a weak antioxidant capacity in their natural state, but their power is remarkably enhanced with the help of peroxidase (POD) [30]. The activity and biosynthesis of POD, an adversity-resistant antioxidase, are induced by ecological stress. Only when the POD activity was high under stress can the components with methoxy groups exert a strong power. These components are usually located at the downstream or terminus of biosynthesised flavonoids and are stable. Therefore, the activities of the components with methoxy groups constantly changed due to ROS formed from ecological stress.

Interestingly, the above-mentioned secondary metabolites were all highly active at relatively low concentrations, indicating that the effect of compounds with a low concentration cannot be ignored.

## Regulation of secondary metabolic pathways

The effects of ROS depend on their concentration in plants, as too much or too little is harmful. Whether there is an overabundance or shortage of ROS depends on the delicate equilibrium between radical production and scavenging [5], which is maintained by antioxidants. Under severe stress, secondary metabolites are highly produced and are coordinated with

**Fig 6. Molecular structure and biosynthesis diagram of flavonoids.** CHI: chalcone isomerase; FNSII-2: flavone synthase; OMT: O-methyltransferases; FH: flavanone hydroxylase; F6H: flavone 6-hydroxylase; F8H: flavone 8-hydroxylase; UBGAT: glucuronyltransferase; BG:β-glucuronidase.

ROS. The more ROS produced, the higher the flavonoid activity is. The flavonoid activity is regulated through the biosynthesis and interconversion of these flavonoids.

When facing environmental stresses, the increased ROS content swiftly triggered a change in the secondary metabolism. First, the glucuronic acid moieties on baicalin and wogonoside

were removed by β-glucuronidase, and baicalin and wogonoside were rapidly converted to baicalein or wogonin as a result of increased activities [31]. Chrysin, with only C-5 and C-7 hydroxyl groups, could also receive a hydroxyl group and was converted into baicalein, norwogonin, and 5,7-dihydroxy-6,8,2',3' -tetramethoxy flavone with more hydroxy groups. The contents of baicalin, wogonoside, and chrysin are all high in the natural state, and furthermore, these reactions are fulfilled by only one step; therefore, the reaction was swiftly performed. Second, ROS can increase the activity of phenylalanine ammonialyase (PAL) [20,32], continuously replenishing flavonoids.

When the ROS content declined, superfluous baicalein or wogonin was converted into baicalin and wogonoside by baicalein7-O-glucuronosyl transferase. The baicalin and wogonoside are confined to certain domains in cells due to hydrophilic glucuronic acid, making it difficult for baicalin and wogonoside to play a role and resulting in a declined antioxidant capacity. In addition, the biosynthesis of flavonoids declined as the level of ROS decreased.

The components with methoxy groups can astutely regulate the antioxidant capacity according to the ROS level, which also plays an important role in maintaining the equilibrium of ROS.

In this paper, only the components with VIP values ≥2 were selected and regarded as the most significant differential chemical markers, and the biosynthesis pathway of 13 different components was summarized, as shown in Fig 6, according to the literature [8,33]. Now, more than 120 components have been reported in *S. baicalensis* [8], and in reality, the number of activity-relevant pathways must be numerous to maintain the equilibrium of ROS as quickly and delicately as possible.

## Conclusion

The diversity of secondary metabolites plays a crucial role in plant bioactivity. A variety of secondary metabolites in *S. baicalensis* was produced in varying proportions. Under ROS, the contents of free flavonoids with many phenolic hydroxyl groups and characteristically high antioxidant activities increased. The change in the proportion and activities of these compounds modified the ability of the plants to eliminate redundant ROS and maintain ROS equilibrium through the biosynthesis and conversion of flavonoids in a manner similar to that occurring in an intricate buffer solution; in addition, the complexity of the active compounds ensures that plants respond to changing environments as quickly and delicately as possible. Herbal medicines are often closely linked to the ecological significance of secondary metabolites, which perfectly explains the diversity and complexity of medicinal plant ingredients.

## Author Contributions

**Supervision:** Ai-hua Zhang.

**Writing – original draft:** Wei Cong, Xiangcai Meng.

**Writing – review & editing:** Ying Shen, Ai-hua Zhang.

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
