## [Decision Letter · Decision Letter 0]

29 May 2020

PONE-D-20-06648

C omplexity of A ctive M edicinical I ngredients  in Radix Scutelariae  with Sodium Hydrosulfite Exposure

PLOS ONE

Dear Dr. Shen,

Thank you for submitting your manuscript to PLOS ONE. After careful consideration, we feel that it has merit but does not fully meet PLOS ONE’s publication criteria as it currently stands. Therefore, we invite you to submit a revised version of the manuscript that addresses the points raised during the review process.

The manuscript has several shortcomings stated in the reviewers' reports which have to be addressed. In addition to this, my advice to the authors is to thoroughly check the text towards improving overall presentation quality and readability.

We look forward to receiving your revised manuscript.

Kind regards,

Branislav T. Šiler, Ph.D.

Academic Editor

PLOS ONE

Reviewers' comments:

Reviewer's Responses to Questions

**Comments to the Author**

1. Is the manuscript technically sound, and do the data support the conclusions?

Reviewer #1: Partly

Reviewer #2: Yes

Reviewer #3: Yes

2. Has the statistical analysis been performed appropriately and rigorously? 

Reviewer #1: N/A

Reviewer #2: No

Reviewer #3: Yes

3. Have the authors made all data underlying the findings in their manuscript fully available?

Reviewer #1: Yes

Reviewer #2: Yes

Reviewer #3: Yes

4. Is the manuscript presented in an intelligible fashion and written in standard English?

Reviewer #1: No

Reviewer #2: No

Reviewer #3: Yes

5. Review Comments to the Author

Reviewer #1: The manuscript entitles “Complexity of Active Medicinical Ingredients in Radix Scutelariae with Sodium Hydrosulfite Exposure” was thoroughly reviewed to evaluate its possible acceptance in PLOS One. Overall, the results and discussion of this study are still weak. The manuscript is poorly prepared and required significant improvement. Therefore, unfortunately, this manuscript may not be suitable for publication in PLOS One.

In order to improve the quality of the manuscript, further specific comments are given below, although the list is not exhaustive:

1. First of all, it is very confusing between Scutellaria baicalensis and Radix Scutelariae as the author used in the title and through the text. At least, the authors should be mention Radix Scutelariae is the root of Scutellaria baicalensis, and used one name in all the text.

2. In the abstract and introduction part, the author mention: “The reason why plants contain so many secondary metabolites is unknown” is not exactly. Although we can not discover all the reason, numerous reason for plants contains many secondary metabolites were observed, such as:

- Some of these chemicals enhance their own survival.

- Some plants produce chemicals that act as herbicides to inhibit the growth of competing plants.

- Other plants produce substances that deter browsing by insects and herbivores

- Many plants produced secondary metabolites to respond to environmental stress (biotic, abiotic stress), etc.

3. It is not necessary and too long for explaining bout ROS and antioxidant enzyme in the introduction part. Better to focus on the main content of these results. The role of Na2S2O4 should be deeply mentioned.

4. In material and method: “… 2-year-old plants of S. baicalensis were sprayed with aqueous Na2S2O4 aqueous solutions at concentrations of 40 μmol/L, 0.4 μmol/L, and 0.004 μmol/L, and the control group was kept moist with just water. The samples were collected 0, 1, 2, and 3 days after spraying”. However, the result was not mentioned about the difference of result among those treatments.

5. Which treatment was showen in Table 1?

6. Result and discussion too short to difficult to understand this study.

7. The authors should establish a metabolic synthesis pathway for this plant. It helps explain the result easily.

8. There have no Fig 5L in Fig. 5.

9. All the species name in the references list should be in Italic.

Reviewer #2: manuscript can be accepted after thorough revision keeping in view following points:

1. English writing.

2. cite relevant references.

3. provide some possible mechanisms in the discussion.

4. please rewrite conclusion and make abstract crispy and to the point.

5. a clear hypothesis is missing.

Reviewer #3: Comments

1. In the Abstract of the article, line 8, it is better for the authors to add the Scutellaria inside the parentheses after the root of Scutellaria, and use it later in the whole article.

2. The authors used the Na2SO4 compound to induce oxidative treatment. What is the reason for using this compound? Explain in the article

3. “Sodium Dithionite-Enhanced Quality of Scutellaria Through Modification of Secondary Metabolism” in the introduction, the last paragraph, the reference should be given.

4. In the materials section, ethanol solvents are used for extraction, while methanol solvents are commonly used to extract flavonoids. What is the reason for use? Explain the reason in manuscript

Results

5. In Table 1, the acronym” m / z “ should be written in Italic

6. In the table in the column related to MS / MS fragment ion, for the compounds that have been identified in both positive and negative ionization methods, the negative ionization pattern is also should be written.

7. In Table 4, combinations 4 and 5 both have the same negative and positive ionization modes, the same m / z, and the same ionization pattern have considered while they did not use any standard combinations.

8. In Table 1, the authors include 25 compounds. Are these compounds first identified from this plant? If not, it is best to briefly explain the composition of each compound and add the relevant references.

9. In Figure 2, the chromatogram images please show the root extract in the positive and negative ionization state of the peak corresponding to each combination with numbers.

Discussion

10. According to statistical analysis, compounds whose VIP is larger or equal to 2 (VIP ≥2) have been selected for further study and the number of these compounds is 13, while in the discussion section of the first paragraph, 12 compounds are mistakenly written. That needs to be edited.

11. In the discussion section of the last paragraph, it is better for the authors to give a brief explanation about the path of biosynthesis of flavonoids in the roots of Scutellaria.

12. In the last paragraph of the last line, in explaining Figure 5, we talk about the multiplicity of biosynthetic pathways of flavonoid compounds, while Figure 5 does not. Figure 5 provides information on the content of compounds identified under different doses of oxidative treatment, and no such connection is made.

13. In Figure 5, it is best to add a chart of the contents of the studied compounds to the image in control mode. The values provided are for treatment only

6. PLOS authors have the option to publish the peer review history of their article (what does this mean?). If published, this will include your full peer review and any attached files.

Reviewer #1: No

Reviewer #2: No

Reviewer #3: Yes: Ali Sharafi

---

## [Author Response · Author response to Decision Letter 0]

3 Jul 2020

Review Comments to the Author

Reviewer #1: The manuscript entitles “Complexity of Active Medicinical Ingredients in Radix Scutelariae with Sodium Hydrosulfite Exposure” was thoroughly reviewed to evaluate its possible acceptance in PLOS One. Overall, the results and discussion of this study are still weak. The manuscript is poorly prepared and required significant improvement. Therefore, unfortunately, this manuscript may not be suitable for publication in PLOS One.

Answer: Thank you for your suggestion. This manuscript aim to clarify why medicine plants contain so many active medicinical ingredients in the perspective of ecological stress. The discussion part were supplemented and throughly revised. 

In order to improve the quality of the manuscript, further specific comments are given below, although the list is not exhaustive:

1. First of all, it is very confusing between Scutellaria baicalensis and Radix Scutelariae as the author used in the title and through the text. At least, the authors should be mention Radix Scutelariae is the root of Scutellaria baicalensis, and used one name in all the text.

Answer: Radix Scutelariae is the root of Scutellaria baicalensis, to avoid confusing, the difference between them was labeled at starting part.

2. In the abstract and introduction part, the author mention: “The reason why plants contain so many secondary metabolites is unknown” is not exactly. Although we can not discover all the reason, numerous reason for plants contains many secondary metabolites were observed, such as:

- Some of these chemicals enhance their own survival.

- Some plants produce chemicals that act as herbicides to inhibit the growth of competing plants.

- Other plants produce substances that deter browsing by insects and herbivores

- Many plants produced secondary metabolites to respond to environmental stress (biotic, abiotic stress), etc.

Answer: It is a problem coming from confussing expression! We mean to express the reason that plants contain so many secondary metabolites, can a few kinds of secondary metabolites with similiar effects do that? Based on this hypothesis, complexity of Active Medicinical Ingredients in Radix Scutelariae can be pesented. The manuscript was revised in another words. 

3. It is not necessary and too long for explaining bout ROS and antioxidant enzyme in the introduction part. Better to focus on the main content of these results. The role of Na2S2O4 should be deeply mentioned.

Answer: This article aim to expound the complexity of active medicinical ingredients in Radix Scutelariae, nevertheless, active medicinical ingredients are often closely linked to the ecological stress, to which ROS and antioxidant enzymes can adjust automatically. It is probably caused by poorly expression, have been revised in article.

4. In material and method: “… 2-year-old plants of S. baicalensis were sprayed with aqueous Na2S2O4 aqueous solutions at concentrations of 40 μmol/L, 0.4 μmol/L, and 0.004 μmol/L, and the control group was kept moist with just water. The samples were collected 0, 1, 2, and 3 days after spraying”. However, the result was not mentioned about the difference of result among those treatments.

Answer: With aqueous Na2S2O4 at different concentrations, Citric acid presented a downtrend, the other ingredient a uptrend, as shown in Fig. 3.

5. Which treatment was showen in Table 1?

Answer: The table. 1 was all the identified components in different treatments, which has been further clarificated.

6. Result and discussion too short to difficult to understand this study.

Answer: A pertinent suggestion! With aqueous Na2S2O4 at different concentrations, The change trends has been added in Result part. The pathway of biosynthesis and transformation of secondary metabolites has been supplemented. 

7. The authors should establish a metabolic synthesis pathway for this plant. It helps explain the result easily.

Answer: Good idea! The pathway of biosynthesis and transformation of secondary metabolites has been supplemented.

8. There have no Fig 5L in Fig. 5.

Answer: it is mistaken written. It has been revised.

9. All the species name in the references list should be in Italic.

 Answer: It has been revised.

Reviewer #2: manuscript can be accepted after thorough revision keeping in view following points:

1. English writing.

Answer: The manuscript has polished by English Editing Services nominated by PLOS ONE, as shown in attachment. If need to further improved, we will contact them. 

2. cite relevant references.

Answer: It has been seriously revised according to PLOS ONE.

3. provide some possible mechanisms in the discussion.

Answer: The discussion part was supplemented many information and throughly revised. The pathway of biosynthesis and transformation of secondary metabolites has been supplemented, and further provided some possible mechanisms.

4. please rewrite conclusion and make abstract crispy and to the point.

Answer: It is a good suggestion and has been revised.

5. a clear hypothesis is missing. 

 Answer: It has been supplemented at last part in Introduction.

Reviewer #3: Comments

1. In the Abstract of the article, line 8, it is better for the authors to add the Scutellaria inside the parentheses after the root of Scutellaria, and use it later in the whole article.

 Answer: Good idea, it has been supplemented. 

2. The authors used the Na2SO4 compound to induce oxidative treatment. What is the reason for using this compound? Explain in the article

Answer: It has been supplemented at last part in Introduction.

3. “Sodium Dithionite-Enhanced Quality of Scutellaria Through Modification of Secondary Metabolism” in the introduction, the last paragraph, the reference should be given.

Answer: It is a good suggestion. It has been supplemented. 

4. In the materials section, ethanol solvents are used for extraction, while methanol solvents are commonly used to extract flavonoids. What is the reason for use? Explain the reason in manuscript

Answer: The methanol solvents is right, the ethyl alcohol was mistakenly written.

Results

5. In Table 1, the acronym” m / z “ should be written in Italic

Answer: All acronym “m / z” in muanuscript has been revised into “m / z” .

6. In the table in the column related to MS/MS fragment ion, for the compounds that have been identified in both positive and negative ionization methods, the negative ionization pattern is also should be written.

Answer: Thank you for your suggestion, we have revised it in the manuscript.

7. In Table 4, combinations 4 and 5 both have the same negative and positive ionization modes, the same m / z, and the same ionization pattern have considered while they did not use any standard combinations.

Answer: The combinations 4 and 5 both have the same negative and positive ionization modes, the same m / z, and the same ionization pattern, they must be isomers. It is reqired standard combinations to identify them, but we have no way to obtain them, so, in article it was revised into “Aspenin-6-C-Arabinose-8-C-glucoside or Chrysin- 6-C- glucose - 8 - C - Araboside”.

8. In Table 1, the authors include 25 compounds. Are these compounds first identified from this plant? If not, it is best to briefly explain the composition of each compound and add the relevant references.

Answer: These compounds first identified are from Scutellaria baicalensis, the relevant references has been add. 

9. In Figure 2, the chromatogram images please show the root extract in the positive and negative ionization state of the peak corresponding to each combination with numbers.

 Answer: Thank you for your suggestion, we have revised it in the manuscript.

Discussion

10.According to statistical analysis, compounds whose VIP is larger or equal to 2 (VIP ≥2) have been selected for further study and the number of these compounds is 13, while in the discussion section of the first paragraph, 12 compounds are mistakenly written. That needs to be edited.

Answer: It was mistaken, has been revised. 

11. In the discussion section of the last paragraph, it is better for the authors to give a brief explanation about the path of biosynthesis of flavonoids in the roots of Scutellaria.

Answer: It is a good suggestion, has been supplemented.

12. In the last paragraph of the last line, in explaining Figure 5, we talk about the multiplicity of biosynthetic pathways of flavonoid compounds, while Figure 5 does not. Figure 5 provides information on the content of compounds identified under different doses of oxidative treatment, and no such connection is made.

Answer: It has been revised.

13. In Figure 5, it is best to add a chart of the contents of the studied compounds to the image in control mode. The values provided are for treatment only

Answer: The values have been provided in Fig.3.

---

## [Decision Letter · Decision Letter 1]

20 Jul 2020

PONE-D-20-06648R1

Complexity of Active Medicinal Ingredients in Radix Scutelariae  with Sodium Hydrosulfite Exposure

PLOS ONE

Dear Dr. Shen,

Thank you for submitting your manuscript to PLOS ONE. After careful consideration, we feel that it has merit but does not fully meet PLOS ONE’s publication criteria as it currently stands. Therefore, we invite you to submit a revised version of the manuscript that addresses the points raised during the review process.

Please see my comments below, under Additional Editor Comments.

We look forward to receiving your revised manuscript.

Kind regards,

Branislav T. Šiler, Ph.D.

Academic Editor

PLOS ONE

Additional Editor Comments (if provided):

Abstract lacks flow. There is no logical connection among sentences. In needs to be rewritten.

The whole text contains vernacular expessions and phrases which cannot be accepted to be published in a scientific journal. Therefore, I strongly encourage the authors to ask a senior scientist to read the manuscript and unbiasedly suggest corrections.

Expressions are randomly capitalized.

Reviewers' comments:

Reviewer's Responses to Questions

**Comments to the Author**

1. If the authors have adequately addressed your comments raised in a previous round of review and you feel that this manuscript is now acceptable for publication, you may indicate that here to bypass the “Comments to the Author” section, enter your conflict of interest statement in the “Confidential to Editor” section, and submit your "Accept" recommendation.

Reviewer #1: All comments have been addressed

Reviewer #3: All comments have been addressed

2. Is the manuscript technically sound, and do the data support the conclusions?

Reviewer #1: Yes

Reviewer #3: Yes

3. Has the statistical analysis been performed appropriately and rigorously? 

Reviewer #1: Yes

Reviewer #3: I Don't Know

4. Have the authors made all data underlying the findings in their manuscript fully available?

Reviewer #1: Yes

Reviewer #3: Yes

5. Is the manuscript presented in an intelligible fashion and written in standard English?

Reviewer #1: No

Reviewer #3: Yes

6. Review Comments to the Author

Reviewer #1: The manuscript was thoroughly reviewed to evaluate its possible acceptance in Plos One. The authors carefully revised and improved all the addressed points commented by the reviewers. The manuscript can be accepted for publication.

Reviewer #3: The authors made their manuscript acceptable for publication and No additional comments are needed.

7. PLOS authors have the option to publish the peer review history of their article (what does this mean?). If published, this will include your full peer review and any attached files.

Reviewer #1: **Yes: **Thanh-Tam Ho

Reviewer #3: No

---

## [Author Response · Author response to Decision Letter 1]

11 Aug 2020

Thank you for your suggestion. The fluidity and logical relationship of the abstract in the manuscript have been revised, and AJE experts have been invited to revise the nonstandard sentences in the whole manuscript, Editing Certificate is included in the attachment. Thank you.

---

## [Editor Report · Decision Letter 2]

13 Aug 2020

PONE-D-20-06648R2

Complexity of Active Medicinal Ingredients in Radix Scutelariae  with Sodium Hydrosulfite Exposure

PLOS ONE

Dear Dr. Shen,

Thank you for submitting your manuscript to PLOS ONE. After careful consideration, we feel that it has merit but does not fully meet PLOS ONE’s publication criteria as it currently stands. Therefore, we invite you to submit a revised version of the manuscript that addresses the points raised during the review process.

The authors are advised to meticulously check the text again in order to rectify vague and non-scientific expressions such as "exposed to either Na2S2O4 or suitable conditions" (Abstract). 

Moreover, the text still contains sentences which lack scientific sense such as "...the most significantly different chemical markers of Na2S2O4 were shikimic acid, citric acid..." (Abstract) - I can hardly believe that sodium hidrosulfite can have chemical markers.

We look forward to receiving your revised manuscript.

Kind regards,

Branislav T. Šiler, Ph.D.

Academic Editor

PLOS ONE

---

## [Author Response · Author response to Decision Letter 2]

24 Aug 2020

Thank you for your suggestion. There are indeed vague and non-scientific expressions in the abstract section of the manuscript. We have revised them. Thank you again for your suggestions.

---

## [Editor Report · Decision Letter 3]

27 Aug 2020

Complexity of Active Medicinal Ingredients in Radix Scutelariae  with Sodium Hydrosulfite Exposure

PONE-D-20-06648R3

Dear Dr. Shen,

We’re pleased to inform you that your manuscript has been judged scientifically suitable for publication and will be formally accepted for publication once it meets all outstanding technical requirements.

Kind regards,

Branislav T. Šiler, Ph.D.

Academic Editor

PLOS ONE
---

## [Editor Report · Acceptance letter]

2 Sep 2020

PONE-D-20-06648R3 

Complexity of Active Medicinal Ingredients in Radix Scutellariae with Sodium Hydrosulfite Exposure 

Dear Dr. Shen:

I'm pleased to inform you that your manuscript has been deemed suitable for publication in PLOS ONE. Congratulations! Your manuscript is now with our production department. 

Kind regards, 

on behalf of

Dr. Branislav T. Šiler 

Academic Editor

PLOS ONE